# Hydrogen Production by Methane Pyrolysis in Molten Cu-Ni-Sn Alloys

David Scheiblehner *[iD], Helmut Antrekowitsch, David Neuschitzer [iD], Stefan Wibner and Andreas Sprung

Department of Nonferrous Metallurgy, Montanuniversitaet Leoben, Franz-Josef-Str. 18, A-8700 Leoben, Austria
* Correspondence: david.scheiblehner@unileoben.ac.at

**Abstract:** Hydrogen is an essential vector for transitioning today's energy system. As a fuel or reactant in critical industrial sectors such as transportation and metallurgy, $H_2$ can diversify the energy mix and supply and provide an opportunity to mitigate greenhouse-gas emissions. The pyrolysis of methane in liquid catalysts represents a promising alternative to producing hydrogen, as its energy demand is comparable to steam methane reforming, and no $CO_2$ is produced in the base reaction. In this work, methane pyrolysis experiments were conducted using a graphite crucible filled with liquid ternary Cu-Ni-Sn alloys at 1160.0 °C. A statistical design of experiments allowed the generation of a model equation that predicts the achievable conversion rates in the ranges of the experiments. Furthermore, the experimental results are evaluated considering densities as well as surface tensions and viscosities in the investigated system, calculated with Butler and KRP equations, respectively. The highest methane conversion rate of 40.15% was achieved utilizing a melt of pure copper. The findings show that a combination of high catalytic activity with a high density and a low viscosity and surface tension of the melt results in a higher hydrogen yield. Furthermore, the autocatalytic effect of pyrolysis carbon is measured.

**Keywords:** hydrogen production; carbon production; methane pyrolysis; liquid catalyst; ternary alloy; surface tension; viscosity; density

## 1. Introduction

In addition to significant increases in the generation and utilization of renewable-based electricity, boosting electrification of end-use sectors, and improving energy efficiency, clean hydrogen and its derivates are essential to diversify the energy mix and supply and decarbonize today's energy system. Therefore, an enormous increase in the demand for $H_2$ is predicted for the coming decades. To achieve this growth, a focus on research into clean technical solutions for producing, storing, transporting, and utilizing hydrogen is crucial for sustaining human prosperity in the 21st century [1–4].

A major part of global hydrogen production is based on steam reforming of methane (SMR), a process with low energy requirements (standard reaction enthalpy of 41.25 kJ/mol $H_2$, calculated in the reaction module of FactSage™ 8.2) and high technical maturity. However, a significant drawback of SMR is carbon dioxide emissions, with more than five kilograms of $CO_2$ released for every kilogram of hydrogen produced, thus exacerbating the greenhouse effect. Water electrolysis offers a possible carbon dioxide-free alternative for hydrogen production. It is important to mention that the advantage of $CO_2$ neutrality only comes into effect when the required energy is provided from renewable sources; considering the high energy intensity of the process—the standard reaction enthalpy is 285.83 kJ/mol $H_2$ (calculated in the reaction module of FactSage™ 8.2)—and the potential growth in the future, hydrogen consumption represents a challenging task [4–12].

This paper investigates another promising option: methane pyrolysis describes the thermal decomposition of $CH_4$ in a nonoxidative environment. Unlike other fossil-fuel-based technologies, methane pyrolysis produces hydrogen and solid carbon and, thus,

without generating carbon oxide emissions (cf. Equation (1)). Furthermore, with a standard reaction enthalpy of 37.44 kJ/mol $H_2$ (calculated in the reaction module of FactSage$^{TM}$ 8.2), the process is significantly less endothermic than water electrolysis [13–20].

$$CH_4 \; \rightarrow \; C + 2\,H_2 \tag{1}$$

The economics of methane pyrolysis strongly depend on natural gas and carbon prices. The annual market demand for carbon black is approximately 16.4 million tons, primarily driven by the rubber industry, e.g., for tire production. Saturating this market with pyrolysis carbon would result in the coproduction of about 5.5 million tons of hydrogen per year. Potential new markets could emerge and, even without economic fields of application, the solid carbon could be stored in significantly smaller volumes than $CO_2$ without the risk of gas leakage in underground reservoirs. Possible disadvantages of this pathway include high capital costs and poor product quality. Pyrolysis hydrogen requires further purification for utilization, e.g., in fuel cells [10,20–24].

A thermodynamic calculation of the equilibrium methane conversion in Equation (1) was performed using the equilib module of FactSage$^{TM}$ 8.2. The results are depicted in Figure 1. At a temperature of 900 °C and a pressure of 1 bar, theoretically, it is possible to decompose 96.4% of the $CH_4$ streaming into the reactor. However, due to kinetic limitations and the high activation energy required to break the stable C-H bonds of $CH_4$ molecules, achieving a reasonable reaction yield is difficult below 1200 °C. Catalysts can enhance the reaction rate at lower temperatures by reducing the activation energy, thus improving the overall economy of the process. Catalytic methane pyrolysis already occurs within a temperature range of 600–900 °C, comparable to the temperature required for steam methane reforming. Various catalysts, including carbonaceous materials or transition metals such as Ni, Co, or Fe, have been the subject of research. As the 3D orbitals of these metals are only partially filled, electrons from the C-H bond can be absorbed, thus efficiently facilitating the decomposition mechanism. Solid metallic catalysts outperform those based on carbon but lose effectiveness at temperatures exceeding 600 °C as pyrolysis carbon deposits at active sites during the reaction. Regeneration, for example, by burning accumulated carbon with air or steam, is possible but results in CO and $CO_2$ emissions and decreased efficiency. One potential solution to mitigate deactivation is the utilization of molten metals and alloys as catalysts where the carbon floats on the liquids because of density differences [16–19,25–29].

Therefore, since the 1990s, research on methane pyrolysis in melts has gained interest. Tin is investigated in various works mainly because of its wide liquid range, low vapor pressure and viscosity at process temperature, low toxicity, and no reactivity with the produced carbon. However, no catalytic activity was measured for pure tin [30–36].

To maximize the process efficiency, further improvement of reaction kinetics is crucial. Therefore, researchers have turned to investigating the efficacy of other pure liquid metals. Wang, Li, et al. [37] measured the catalytic activity of magnesium not only for the decomposition of methane but also ethane, polyolefins (plastic and rubber), and asphalt. Zeng, Tarazkar, et al. [38] investigated methane pyrolysis in the presence of liquid and gaseous tellurium, respectively. Due to its high electron affinity, Te was identified as an effective catalyst, significantly lowering the activation energy of the reaction. Leal Pérez, Medrano Jiménez, et al. [39] conducted experiments with liquid gallium and could achieve a $CH_4$ conversion rate of 91% at 1119 °C [33,37–39].

Upham, Agarwal, et al. [40] stated that metals characterized as effective catalysts, such as nickel, platinum, or palladium, generally have high melting points, while low-melting metals, such as indium, bismuth, tin, gallium, or lead, exhibit poor catalytic activity. The concept of mixing elements of those two classifications to create catalytic alloys with economic liquidus temperatures has been explored. The authors reported that a liquid consisting of 27% nickel in bismuth achieved a methane conversion of 95% at 1065 °C. Rahimi, Kang, et al. [41] investigated the same alloy and reduced metal contamination in the produced carbon by adding a molten salt layer. Palmer, Tarazkar, et al. [42] observed

a catalytic potential in mixtures of bismuth and copper, despite the inactivity of the individual constituent metals. Certain copper–bismuth alloys exhibited superior activity compared to Ni27Bi73. Zaghloul, Kodama, et al. [36] investigated liquid tin–copper and tin–nickel systems and observed improved methane conversion rates compared to pure tin. Scheiblehner et al. [43] analyzed various binary copper alloys and concluded that high catalytic activity and reduced surface tension, resulting in smaller gas bubbles, significantly enhanced reaction yield [36,40–43].

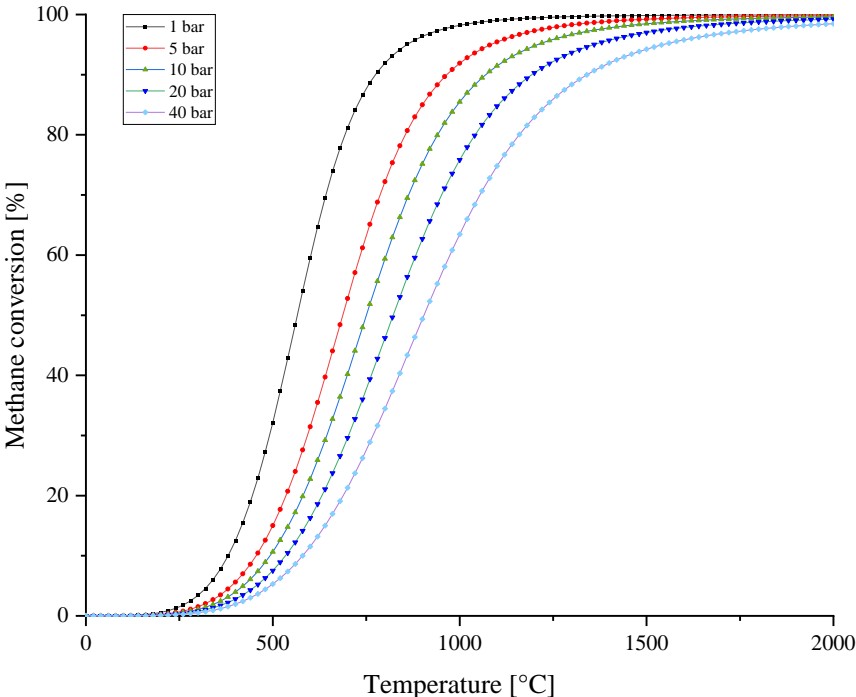

**Figure 1.** Equilibrium $CH_4$ conversion in methane pyrolysis as a function of the temperature at different pressures (data from FactSage$^{TM}$ 8.2).

In addition to increasing the process temperature and catalytic potential of the used liquid media, the following factors were proven to increase the $CH_4$ conversion rate [31–35,41,43]:

- Maximizing gas residence times in the reaction zone, realized, e.g., by increasing the height of the liquid metal column [31–35,41];
- Decreasing the volumetric input flow of methane [32–35,41];
- Generating small, homogenously distributed gas bubbles to maximize the reactive surface, thus enhancing mass and heat transfer rates [31,43].

The dominant aggregate used in the abovementioned works is the bubble column reactor, where highly complex fluid dynamic interactions influence the reactions. Zahedi, Saleh, et al. [44] demonstrated that an increased surface tension leads to longer detachment times of bubbles from the orifice. Therefore, more gas enters the bubble, increasing its diameter. Higher liquid densities were found to result in smaller bubble diameters. These relationships are reflected in empirical formulas that describe the formation of bubbles at single nozzles in different melts, as developed among others in the works of Tate [45], Mori, Sano, et al. [46], and Sano and Mori [47]. Furthermore, liquid viscosity influences hydrodynamics and bubble behavior. Using carboxyl methyl cellulose to change the viscosity of a water–air system, Wu, Wang, et al. [48] measured, that after exceeding a critical value, an increase in viscosity leads to larger bubble diameters. In contrast, the gas holdup presents the opposite variation. Besagni, Inzoli, et al. [49] associated lower viscosities with the formation of more and finer bubbles, promoting a stable flow regime and increasing gas holdup. Mouza, Dalakoglou, et al. [50] demonstrated that an increase in liquid viscosity leads to reduced turbulence in the liquid. This effect enhances bubble

coalescence and diminishes bubble breakage, eventually leading to the formation of larger bubbles [44–50].

This work investigates the effectiveness of various ternary Cu-Ni-Sn alloys as heat transferring and catalytic media in methane pyrolysis. A statistical experimental design enables the generation of a model equation applicable to a specific area of the ternary system. Furthermore, the alloys' density, surface tension, and viscosity are calculated and compared with data from the literature and evaluated regarding the experimental findings. The results provide information on the critical parameters influencing the reaction yield and give insight into possibilities to improve the overall process efficiency by altering the content of alloying elements.

## 2. Materials and Methods

### 2.1. Experimental Setup

A schematic representation of the experimental setup is depicted in Figure 2. The reaction vessel (graphite crucible, $d_i$ = 65 mm, $d_o$ = 95 mm), containing the liquid metallic catalyst, is connected to a piping system to generate a hermetically sealed system. An induction coil heats the reaction zone. After reaching the process temperature, an alumina lance ($Al_2O_3$ > 99.5 wt%; $d_o$ = 8 mm) with 6 holes ($d_i$ = 0.8 mm) is lowered into the melt until the distance between the lower end of the lance and the inner crucible bottom is 5 mm. $CH_4$ with a purity of >99.5 vol% is introduced through the $Al_2O_3$ lance at a volumetric flow rate of 0.5 L/min and bubbles through the liquid metal, where methane decomposition occurs. Nitrogen is injected into the gas space above the melt (flow rate of 1 L/min) through a steel lance to cool this section and suppress unwanted reactions. The $N_2$ flushing furthermore increases the gas volume flow and prevents the accumulation of solid particles in the piping section. Pyrolysis carbon floats on the melt or is discharged in the produced gas stream. The off-gases leave the setup through a hot gas filter (Pall Dia-Schumalith® DS 03e20 filter element; $d_o$ = 50 mm, $d_i$ = 20 mm, h = 135 mm) that separates residual solid particles collected in a glass container. To avoid the condensation of possibly formed polyaromatic hydrocarbons (PAHs) in the filter, external heating ($\vartheta \leq 400\ °C$) is installed. The cleaned exhaust gases are analyzed in an ABB EL 3020 with an Uras26 infrared photometer and Caldos27 thermal conductivity analyzer. A thermocouple (type K) on the outside crucible wall allows continuous temperature monitoring during the experiments.

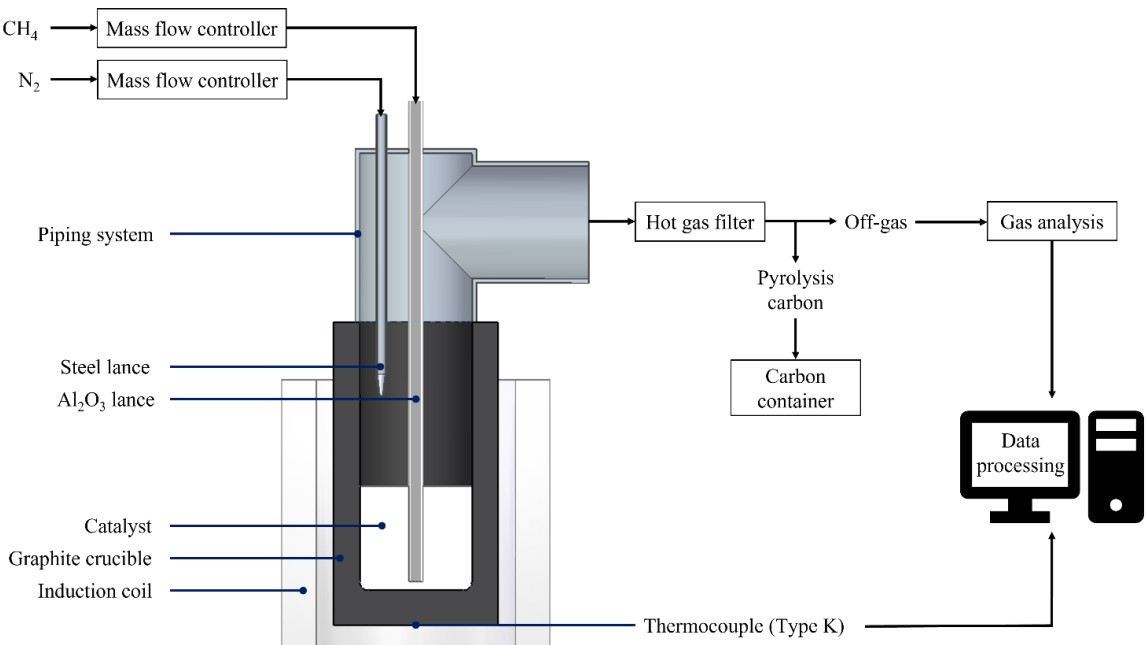

**Figure 2.** Schematic representation of the experimental setup.

### 2.2. Investigated Metals and Alloys

According to a statistical experimental plan (developed using the software MODDE$^{\circledR}$ 12.1; cf. Table 1), 20 alloys have been produced by mixing copper chips (Cu > 99.90 wt%), tin ingots (Sn > 99.99 wt%), and nickel shots (Ni > 99.99 wt%). Runs 17–20 represent center points in the design of experiments. The bath height was defined at 70 mm, as former experiments had shown that this configuration results in an acceptably low amount of ejected catalyst while enabling the gathering of representative data. The liquidus temperature of each sample at 1 bar is calculated with the equilib module of FactSage$^{\text{TM}}$ 8.2. The highest value is computed for CuNi10 with 1142.7 °C. Therefore, the process temperature is defined at 1160.0 °C. The liquidus temperatures of the different metals and alloys, respectively, are listed in Table 1.

**Table 1.** Investigated metals and alloys (element contents in at%) and their liquidus temperatures at 1 bar computed in the equilib module of FactSage$^{\text{TM}}$ 8.2.

| Experiment Nr. | Metal/Alloy | Liquidus Temperature [°C] |
|:---:|:---:|:---:|
| 1 | CuSn33.334Ni3.334 | 722.50 |
| 2 | CuSn66.665Ni3.334 | 494.27 |
| 3 | CuSn33.335Ni6.667 | 750.41 |
| 4 | CuSn66.665Ni6.667 | 567.19 |
| 5 | CuSn16.665Ni1.667 | 811.09 |
| 6 | CuSn83.335Ni1.667 | 436.84 |
| 7 | CuSn16.665Ni8.334 | 835.39 |
| 8 | CuSn83.335Ni8.334 | 775.47 |
| 9 | Cu | 1084.62 |
| 10 | Sn | 231.92 |
| 11 | CuNi10 | 1142.70 |
| 12 | SnNi10 | 904.02 |
| 13 | CuNi3.333 | 1104.80 |
| 14 | SnNi6.667 | 875.51 |
| 15 | CuSn33.333 | 701.30 |
| 16 | CuSn66.667Ni10 | 657.57 |
| 17 | CuSn50Ni5 | 648.76 |
| 18 | CuSn50Ni5 | 648.76 |
| 19 | CuSn50Ni5 | 648.76 |
| 20 | CuSn50Ni5 | 648.76 |

## 3. Calculations

The density of each alloy, $\rho_{\text{alloy}}$, consisting of N metals at a given temperature, is approximated in Equation (2), with $w_i$ representing the mass fraction and $\rho_i$ the density of pure a component, i, at a given temperature (data acquired from Gale and Totemeier [10]), respectively.

$$\rho_{\text{alloy}} = \frac{1}{\sum_{i=1}^{N} \frac{w_i}{\rho_i}} \tag{2}$$

The required mass of each metal, $m_i$, is calculated using Equation (3). $V_{\text{alloy}}$ represents the bath volume, equal to 0.232 L in the present experimental setup.

$$m_i = \rho_{\text{alloy}} \cdot V_{\text{alloy}} \cdot w_i \tag{3}$$

Figure 3 depicts the calculated density in the liquid area of the ternary system Cu-Ni-Sn at a temperature of 1160.0 °C and a pressure of 1 bar. The results for the 20 investigated metals and alloys are listed in Table 2.

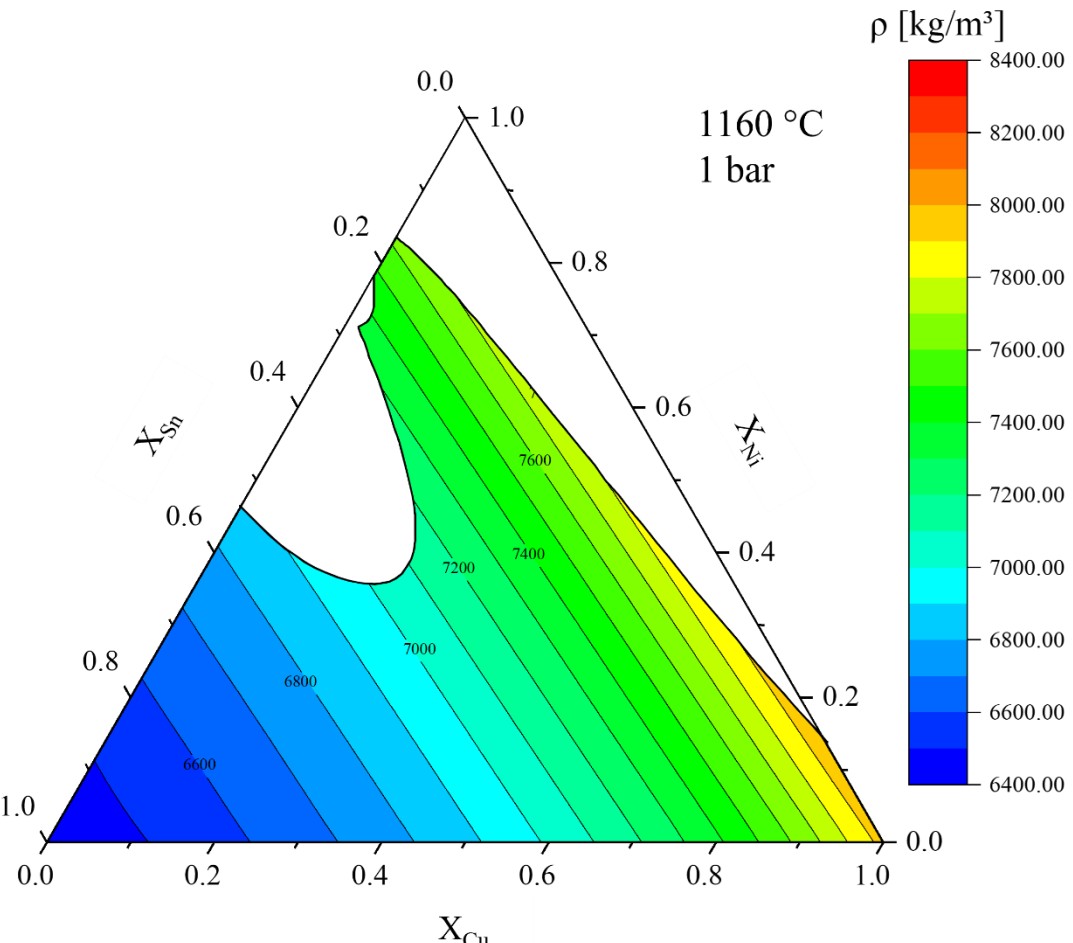

**Figure 3.** Density in the system Cu-Ni-Sn at 1160.0 °C and 1 bar (liquidus lines computed with FactSage™ 8.2).

Butler [51] proposed a thermodynamic equilibrium between the bulk phase and a monoatomic surface layer of a molten alloy to estimate its surface tension, σ, as given in Equation (4). Here, $\sigma_i$ is the surface tension and $A_i$ is the partial molar surface area of the pure component i. The surface tensions of pure metals at process temperature are obtained from Gale and Totemeier [10]. $X_i^{sup}$ describes the molar fraction and $G_i^{E, sup}$ the partial excess Gibbs energy of component i in the surface phase (sup = S) or the bulk phase (sup = B), respectively. R is the molar gas constant and T is the temperature.

$$\sigma = \sigma_i + \frac{RT}{A_i} \ln\frac{X_i^S}{X_i^B} + \frac{1}{A_i}\left[G_i^{E, S}\left(T, X_i^S\right) - G_i^{E, B}\left(T, X_i^B\right)\right] \tag{4}$$

For practical purposes, $A_i$ is determined according to Equation (5) where L is a geometric factor (L = 1.091 for molten alloys according to Tanaka and Iida [52]), $V_{m, i}$ is the partial molar volume of component i and $N_{Av}$ is the Avogadro number. The partial molar volume of component i is determined using data from Gale and Totemeier [10].

$$A_i = L \cdot V_{m, i}^{2/3} \cdot N_{Av}^{1/3} \tag{5}$$

The partial excess Gibbs energy of component i in the surface layer can be obtained from an estimation proposed by Speiser, Poirier et al. [53] (cf. Equation (6)) where the coefficient β describes the ratio of the coordination number in the surface phase to the coordination number in the bulk phase. For this work's calculations, β is defined at 0.83, as Tanaka, Hack, et al. [54] recommended for liquid alloys. The excess Gibbs free energy of

the bulk phase of the sub-binary and ternary systems is approximated using Redlich–Kister polynomials. The necessary thermodynamic data were obtained from Zhao, Yang, et al. [55], Gnanasekaran and Ipser [56], Li, Guo, et al. [57], and Li, Franke, et al. [58].

$$G_i^{E, S}\left(T, X_i^S\right)= \beta \cdot G_i^{E, B}\left(T, X_i^B\right) \tag{6}$$

By substituting Equations (5) and (6) in Equation (4), the surface tensions of the investigated ternary alloys are determined. Figure 4 depicts the calculated values at a temperature of 1160.0 °C and a pressure of 1 bar. The calculated surface tensions of the 20 investigated metals and alloys are listed in Table 2.

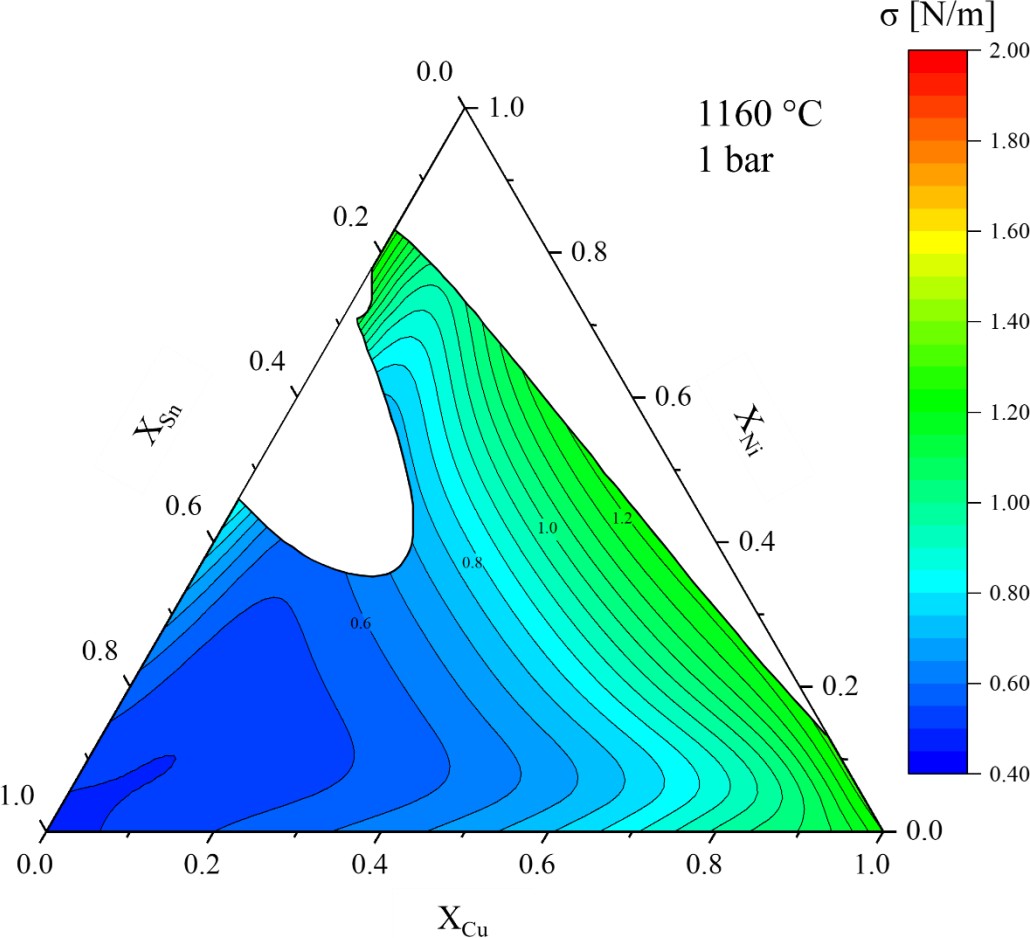

**Figure 4.** Surface tension in the system Cu-Ni-Sn at 1160.0 °C and 1 bar calculated with Butler equation [51] (liquidus lines computed with FactSage™ 8.2).

The Kozlov–Romanov–Petrov (KRP) equation (cf. Equation (7)) developed by Kozlov, Romanov, et al. [59] represents a possibility to approximate the viscosity of an alloy, $\eta$, based on the viscosities of the pure components, $\eta_i$, and their mixing enthalpy, $\Delta H$. Viscosities of the pure metals at 1160.0 °C are acquired from Gale and Totemeier [10], and the mixing enthalpies are determined using data from Zhao, Yang, et al. [55].

$$Ln\eta = \sum_{i=1}^{N} x_i \cdot \ln \eta_i - \frac{\Delta H}{3 \cdot R \cdot T} \tag{7}$$

Figure 5 depicts the computed values for the investigated ternary system. The calculated viscosities of the 20 investigated metals and alloys are listed in Table 2.

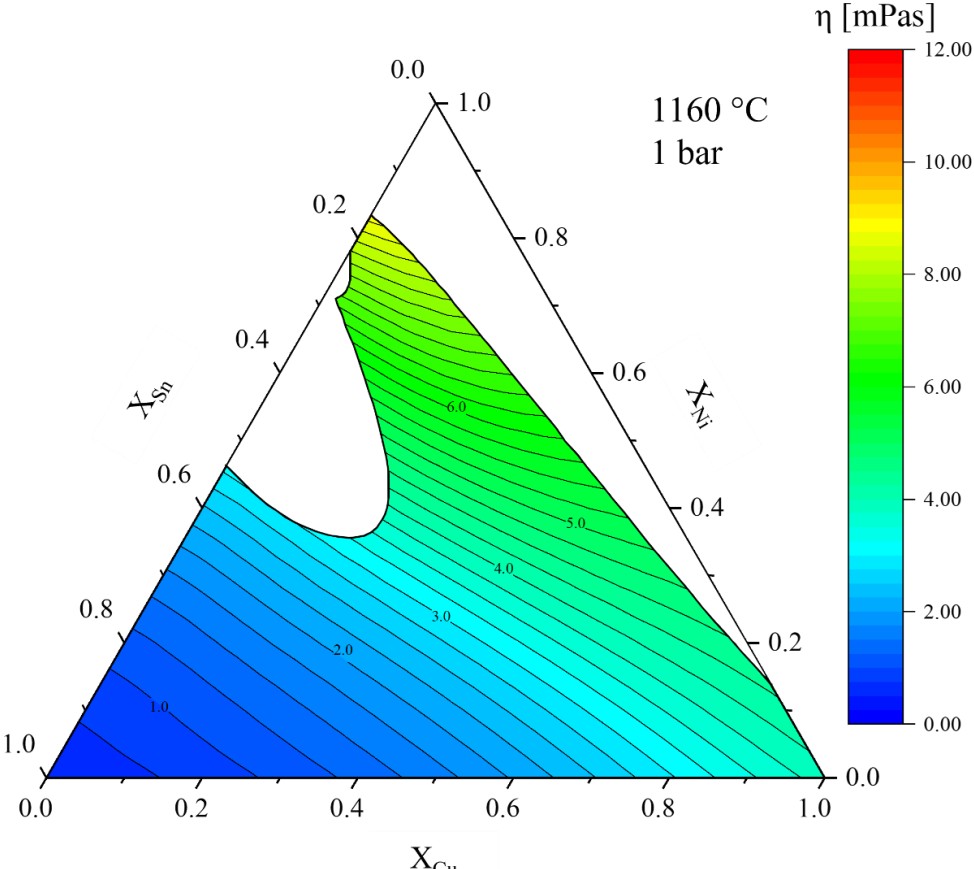

**Figure 5.** Viscosity in the system Cu-Ni-Sn at 1160.0 °C and 1 bar calculated with KRP equation [59] (liquidus lines computed with FactSage™ 8.2).

**Table 2.** Approximated densities, surface tensions (calculated with Butler equation [51]), viscosities (calculated with KRP equation [59]), and $CH_4$ conversion rates after time increments of 30, 40, 50, and 60 min of the investigated metals and alloys at 1160.0 °C and 1. bar.

| Experiment Nr. | Density [kg/m³] | Surface Tension [N/m] | Viscosity [mPas] | $CH_4$ Conversion Rate [%] after t [min] | | | | |
|---|---|---|---|---|---|---|---|---|
| | | | | t = 20 | t = 30 | t = 40 | t = 50 | t = 60 |
| 1 | 7125.28 | 0.67 | 2.02 | 20.15 | 20.17 | 21.41 | 22.25 | 23.14 |
| 2 | 6687.40 | 0.56 | 1.10 | 20.32 | 20.23 | 20.70 | 21.16 | 21.40 |
| 3 | 7129.70 | 0.69 | 2.09 | 15.43 | 15.75 | 15.95 | 16.52 | 16.83 |
| 4 | 6689.92 | 0.53 | 1.14 | 19.32 | 20.85 | 20.55 | 20.48 | 21.25 |
| 5 | 7459.40 | 0.79 | 2.56 | 23.73 | 28.82 | 30.76 | 31.71 | 32.91 |
| 6 | 6536.17 | 0.52 | 0.78 | 22.48 | 23.07 | 23.38 | 23.08 | 23.99 |
| 7 | 7471.94 | 0.86 | 2.71 | 23.44 | 17.79 | 19.55 | 20.35 | 22.19 |
| 8 | 6540.15 | 0.50 | 0.84 | 22.34 | 23.17 | 22.73 | 23.72 | 23.39 |
| 9 | 7938.28 | 1.29 | 3.55 | 15.24 | 20.47 | 35.16 | 40.15 | 40.06 |
| 10 | 6413.83 | 0.48 | 0.68 | 26.65 | 28.95 | 30.94 | 32.82 | 35.56 |
| 11 | 7966.70 | 1.28 | 3.79 | 16.61 | 17.10 | 17.41 | 16.80 | 19.52 |
| 12 | 6489.19 | 0.52 | 0.88 | 21.77 | 22.32 | 23.37 | 25.63 | 27.96 |
| 13 | 7947.68 | 1.28 | 3.63 | 31.43 | 34.46 | 34.32 | 33.84 | 33.53 |
| 14 | 6462.99 | 0.50 | 0.81 | 28.19 | 30.01 | 29.47 | 29.28 | 30.42 |
| 15 | 7120.89 | 0.81 | 2.05 | 16.38 | 17.60 | 17.98 | 17.75 | 18.47 |
| 16 | 6692.44 | 0.53 | 1.19 | 17.15 | 17.39 | 17.44 | 18.40 | 19.12 |
| 17 | 6879.00 | 0.59 | 1.53 | 17.26 | 17.48 | 17.73 | 18.52 | 17.95 |
| 18 | 6879.00 | 0.59 | 1.53 | 15.23 | 15.50 | 16.78 | 17.09 | 16.01 |
| 19 | 6879.00 | 0.59 | 1.53 | 18.17 | 18.69 | 18.99 | 18.34 | 17.77 |
| 20 | 6879.00 | 0.59 | 1.53 | 15.29 | 15.76 | 15.49 | 15.79 | 15.90 |

## 4. Results and Discussion

During the experiments, the exhaust gas is analyzed continuously to determine the content of hydrogen, $X_{H_2}$, which can be calculated according to Equation (8), with $Q_{H_2}$, $Q_{CH_4,o}$, and $Q_{N_2}$ representing the volume flow of produced hydrogen, outflowing $CH_4$, and inert $N_2$, respectively.

$$X_{H_2} = \frac{Q_{H_2}}{Q_{H_2} + Q_{CH_4,o} + Q_{N_2}} \cdot 100\% \tag{8}$$

Expressing $Q_{H2}$, $Q_{CH4,\,o}$, and $Q_{N2}$ as functions of the conversion rate of methane, $CR_{CH4}$, and the input volume flow of methane, and $Q_{CH4,i}$ results in the formulation of Equation (9).

$$X_{H_2} = \frac{Q_{CH_4,i} \cdot \dfrac{CR_{CH_4}}{100\%} \cdot 2}{Q_{CH_4,i} \cdot \dfrac{CR_{CH_4}}{100\%} \cdot 2 + Q_{CH_4,i} \cdot \left(1 - \dfrac{CR_{CH_4}}{100\%}\right) + Q_{N_2}} \cdot 100\% \tag{9}$$

Transcribing Equation (9) gives an expression that enables the computation of $CR_{CH4}$ solely from input parameters or measured values (cf. Equation (10)). Table 2 lists the results after time increments of 30, 40, 50, and 60 min.

$$CR_{CH_4} = \frac{X_{H_2} \cdot \left(Q_{CH_4,i} + Q_{N_2}\right)}{Q_{CH_4,i} \cdot \left(2 - \dfrac{X_{H_2}}{100\%}\right)} \tag{10}$$

Employing the statistical software MODDE® 12.1, it is possible to describe the measured values with mathematical functions by multiple linear regression. Thus, coefficients are defined, which can be used to specify an applicable equation within the range of alloy compositions investigated in this work (cf. Equation (11), with coefficients $K_1$–$K_6$ and $X_{Sn}$ and $X_{Ni}$ representing the molar fraction of Sn and Ni, respectively). The resulting equation can predict the achievable $CH_4$ conversion rate for a given catalyst composition and experimental time. The determined coefficients for an experimental time of 20 and 40 min, respectively, are listed in Table 3.

$$CR_{CH_4} = K_1 + X_{Sn} \cdot K_2 + X_{Ni} \cdot K_3 + X_{Sn}^2 \cdot K_4 + X_{Ni}^2 \cdot K_5 + X_{Sn} \cdot X_{Ni} \cdot K_6 \tag{11}$$

**Table 3.** Coefficients for the calculation of the $CH_4$ conversion rate after an experimental time of 20 and 40 min, respectively.

| Coefficient | Value after 20 Min | Value after 40 Min |
|:---:|:---:|:---:|
| $K_1$ | 30.259 | 36.029 |
| $K_2$ | −0.554 | −0.657 |
| $K_3$ | 0.643 | −0.113 |
| $K_4$ | 0.005 | 0.006 |
| $K_5$ | −0.168 | −0.155 |
| $K_6$ | 0.013 | 0.019 |

The results for an experimental time of 20 and 40 min are visualized in Figures 6 and 7, respectively.

For most investigated alloys, the methane conversion rate tends to improve with the experimental time (cf. Table 2). This effect is attributed to generated pyrolysis carbon that forms a floating layer on the melt or is deposited on reactor walls and in the pipes and also has a catalytic effect to some extent (autocatalysis) [60,61].

The literature research has shown that effects enhancing the reaction yield in bubble column reactors can be correlated with lower surface tensions and viscosities as well as

higher densities of the utilized liquids [42–50]. The experimental results are evaluated considering a superposition of these parameters.

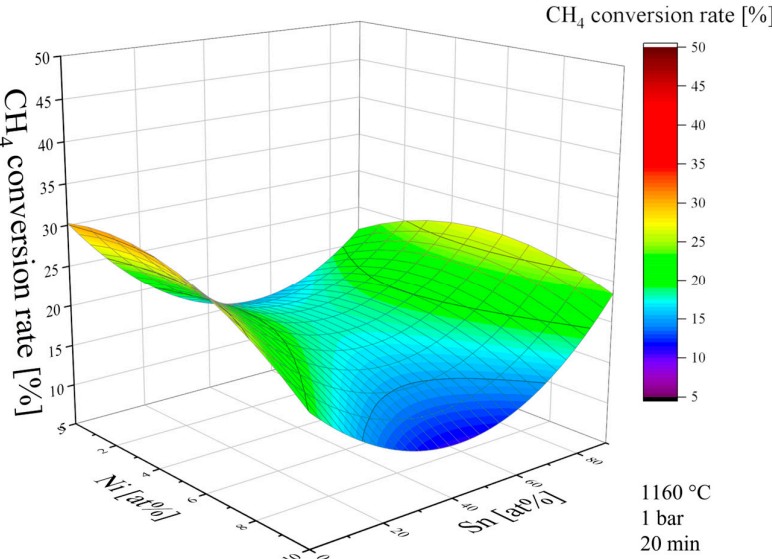

**Figure 6.** $CH_4$ conversion rate in a Cu-Ni-Sn alloy after an experimental time of 20 min as a function of Sn and Ni content, respectively (calculated using coefficients determined in MODDE® 12.1).

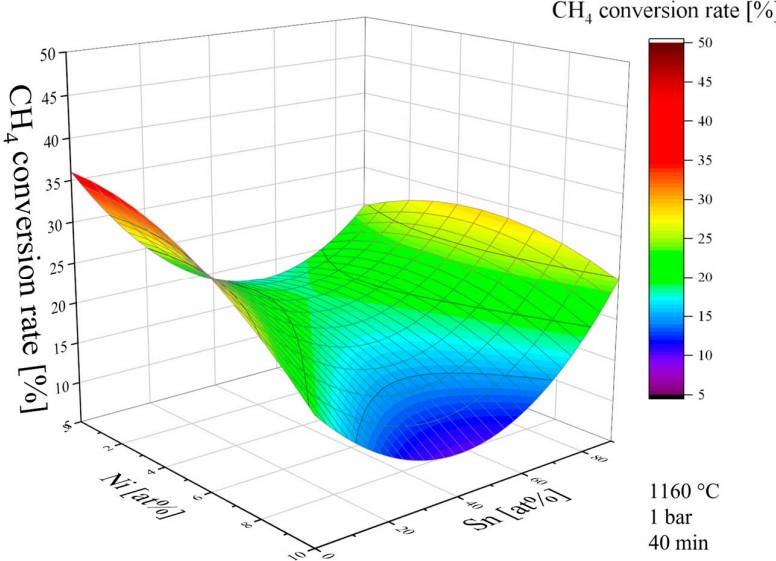

**Figure 7.** $CH_4$ conversion rate in a Cu-Ni-Sn alloy after an experimental time of 40 min as a function of Sn and Ni content, respectively (calculated using coefficients determined in MODDE® 12.1).

Nickel shows a high catalytic activity in methane pyrolysis [62,63]. Therefore, it is expected to positively impact the CH4 conversion in the investigated process. However, in this work's experiments, small quantities of Ni in the system Cu-Sn tend to have a minor beneficial impact, if any, while adding higher amounts of nickel has a detrimental effect on the reaction yield. The calculations indicate that gradually substituting copper with nickel in the investigated range of the ternary system Cu-Ni-Sn results in a slightly increasing density (cf. Figure 3) and a rise in viscosity (cf. Figure 5). On the other hand, adding small amounts of nickel leads to a minor decrease in the computed surface tension until a minimum is reached (cf. Figure 4). Further addition of nickel increases the calculated values marginally. These findings indicate that the beneficial effects of high catalytic activity, in combination with an increasing density and decreasing surface tension, compensate for the

adverse impact of a rising viscosity at a low nickel content. However, as more Ni is added, detrimental effects, such as increased surface tension and viscosity, prevail, and the $CH_4$ conversion rate declines.

In previous investigations, no catalytic effect was determined for pure tin [32–35]. This work measures a declining conversion rate as Sn is mixed with Cu-Ni. A minimum is reached between 35–60% Sn, depending on Ni content and experimental time (cf. Figures 6 and 7). A further increase in the content of tin results in a gradual improvement of methane conversion. According to these works' calculations, tin lowers the investigated alloys' density (cf. Figure 3), surface tension (cf. Figure 4), and viscosity (cf. Figure 5). Therefore, the decrease in the methane conversion rate at a low Sn content can be attributed to detrimental effects due to a decreasing density and the substitution of possibly catalytic copper—Zaghloul, Kodama, et al. [36] showed that Cu-Sn alloys achieved higher reaction rates than pure Sn—with noncatalytic tin. Furthermore, the viscosity in the binary system Cu-Sn was investigated by Tan, Xiufang, et al. [64]. In their measurements, viscosity reached a maximum between 10 and 25% Sn. This behavior could not be depicted in this work's calculations but provides another possible explanation for the measured results. As more and more tin is added, the beneficial effects of low surface tension and viscosity prevail, leading to a reincrease in the $CH_4$ conversion rate.

At approximately 40%, the highest methane conversion rates were obtained using pure copper at the end of the experiments. This observation underlines the assumption that copper shows high catalytic activity in methane pyrolysis. Investigating a metal bath of pure tin, the highest achieved $CH_4$ conversion rate was almost 36%. Geißler, Abánades, et al. [34] measured a methane conversion of 78% at 1175 °C using pure Sn. In their work, methane is introduced at the bottom of a Sn column with a height of 1050 mm (measured at 1000 °C). Additionally, the reactor is filled with cylindrical quartz glass rings, forming a packed bed with an average porosity of 84 vol-%. Thus, the high conversion rates compared to the present results can be attributed to the residence time of the generated gas bubbles, which is many times higher than that of the gas bubbles formed in our reactor. Furthermore, the difference in the reaction temperature has an influence, even if only a minor one. In the works of Zaghloul Kodama, et al. [36], using pure tin resulted in a conversion rate of 12% with a 10 cm molten-metal column at 1050 °C. Although the volume flow of input gas in their work is relatively low (0.07 L/min), the measured methane conversion is lower than ours. This observation is attributable mainly to the difference in experimental temperature.

In addition, the following issues regarding materials and methods must be addressed:

- One of the main advantages of the utilized reactor is its simplicity which results in low maintenance requirements while enabling the investigation of a wide range of different metals and alloys with sufficient accuracy and quick changes of the input material. For similar reasons, methane is introduced through a submerging lance and not via devices that would lead to a more homogenous distribution of smaller bubbles, such as bottom flushers or impellers;

- Methane dissociation occurs not only in the melt but also in the reactor headspace above. Although the installed nitrogen flushing cools the gas section above the liquid, thus suppressing unwanted reactions and distortion of the results, the actual contribution of methane dissociation in the gas phase to the total conversion rate is yet to be determined. It requires more detailed monitoring of the temperature profile over the reactor height. However, as the share of this headspace conversion is expected to be similar for all investigated metals and alloys, a comparison is still possible with the current experimental setup;

- An ideal behavior of the melt is assumed to calculate the alloy densities according to Equation (2). Thus, volumetric changes due to attracting and repelling forces between different metal atoms are not considered. Although this discrepancy impacts the exact bath height at process conditions, previous experiments have shown that the approximated values are satisfactory to obtain comparable results;

- No thermocouple is placed inside the melt in this work's experiments, as prior investigations indicated that the deviation to the outside of the graphite crucible is negligible, mainly due to the high thermal conductivity of the metallic catalyst and the crucible.

## 5. Conclusions

Various scenarios predict an enormous increase in hydrogen demand over the next decade, requiring sharply intensified research efforts into technical solutions for $CO_2$-neutral production. Methane pyrolysis is a promising approach as no carbon dioxide emissions are produced in the base reaction, and its energy consumption is comparable to traditional, fossil-fuel-based methods such as, e.g., SMR.

This work examines the production of hydrogen and solid carbon via the thermocatalytic decomposition of methane. Therefore, mixtures of copper, nickel, and tin were used as catalytic and heat-transferring media. The investigated ternary system's calculated viscosities and surface tensions are compared with the literature results. A statistical evaluation of the measured $CH_4$ conversion, in combination with the findings of these computations, facilitates the identification of beneficial properties and parameters.

The autocatalytic effect of generated pyrolysis carbon is measured. A combination of high catalytic activity with a low viscosity and surface tension and high density of the melt results in a maximum hydrogen yield. The highest methane conversion rate of 40% could be achieved at the end of the experimental time using pure copper as the catalytic, heat transferring at 1160 °C, and atmospheric pressure. The highest thermodynamically achievable conversion rate at this temperature and pressure is 99.31%. Without changing the parameters mentioned, a gas-injection system that generates a homogenous distribution of finer bubbles and an increased bath height resulting in maximum residence times may facilitate better hydrogen yields. However, the aim of this work is the investigation of the influence of different metals and alloys. For this purpose, this work's experimental setup is favorable.

The calculated surface tensions and viscosities must be evaluated and compared to practical results. However, due to the lack of experimental data to characterize the ternary system's relevant properties precisely, conclusions can only be drawn conditionally, highlighting the importance of additional research in this area.

**Author Contributions:** Methodology, D.S.; Validation, D.S.; Investigation, D.S., D.N. and A.S.; Resources, A.S.; Writing—original draft, D.S.; Writing—review & editing, H.A., D.N. and S.W.; Supervision, H.A. and S.W.; Project administration, H.A. and S.W. All authors have read and agreed to the published version of the manuscript.

**Funding:** This research received no external funding.

**Data Availability Statement:** The data presented in this study are available on request from the corresponding author.

**Conflicts of Interest:** The authors declare no conflict of interest.

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
