# Peer review of "Hydrogen Production by Methane Pyrolysis in Molten Cu-Ni-Sn Alloys"

_metals, doi:10.3390/met13071310_

Round 1

Reviewer 1 Report

The researchers investigated the effects of density, viscosity, and surface tension on catalytic hydrogen production when using molten alloys as catalysts. However, the paper did not propose the best experimental conditions and the best conversion rate that can be achieved. The author should increase the amount of data, propose the best alloy composition and the best experimentalconditions, and compare with the work of others. The work is incomplete,and the values of density, viscosity and surface tension are theoreticalcalculations and lack of experimental characterization. Therefore, I do not recommend the publication of this article.

Extensive editing of English language required.

Reviewer 2 Report

The present work examines the production of hydrogen and solid carbon via the thermo-catalytic decomposition of methane using mixtures of copper, nickel, and tin at 1160.0 °C as catalytic and heat-transferring media.  A statistical design of experiments allowed the generation of a model equation that predicts the achievable conversion rates in the ranges of the suggested conditions. As the main conclusions of this research are that, the autocatalytic effect of generated pyrolysis carbon is measured. Moreover, a combination of high catalytic activity with a low viscosity and surface tension and high density of the melt results in a maximum hydrogen yield are suggested.

Author Response

Thank you very much for the constructive feedback.

Reviewer 3 Report

The authors investigated the effectiveness of various ternary Cu-Ni-Sn alloy as heat transferring and catalytic media in methane pyrolysis. The alloy's density, surface tension and viscosity were also caluculated. The methane conversion is not very high in the system, but those data may be useful to readers. However, there are some questions as follows.

・In reference [29], the methane conversion rate was 78% at 1175°C using Sn metal, but in this system, the methane conversion rate was about 31% even if the pyrolysis was performed at 1160°C using Sn. Why are the values for this system lower than those in the reference?

・The authors stated that the methane conversion rate of pure Cu was high because pure Cu had catalytic activity, but the conversion rate is 40.06%, which does not seem to be a higher value compared to the data in some references. Is there evidence for the existence of a catalytic effect in that system?

・In the case of alloys containing Ni, is there no catalytic effect of Ni?

Round 2

Reviewer 1 Report

I recommend this article publication.

Minor editing of English language required